# Nutritional Risk Screening Tools for Older Adults with COVID-19: A Systematic Review

**DOI:** 10.3390/nu12102956

**Published:** 2020-09-27

**Authors:** David Franciole Oliveira Silva, Severina Carla Vieira Cunha Lima, Karine Cavalcanti Mauricio Sena-Evangelista, Dirce Maria Marchioni, Ricardo Ney Cobucci, Fábia Barbosa de Andrade

**Affiliations:** 1Postgraduate Program in Collective Health, Federal University of Rio Grande do Norte—UFRN, Natal 59056-000, Brazil; 2Department of Nutrition, Federal University of Rio Grande do Norte—UFRN, Natal 59078-970, Brazil; scvclima@gmail.com (S.C.V.C.L.); kcmsena@gmail.com (K.C.M.S.-E.); 3Department of Nutrition, School of Public Health, University of São Paulo—USP, São Paulo 05410-020, Brazil; marchioni@usp.br; 4Postgraduate Program in Biotechnology, Potiguar University—UnP, Natal 59056-000, Brazil; drcobucci@ufrn.edu.br

**Keywords:** nutritional screening, nutritional risk, nutritional assessment, malnutrition, elderly, COVID-19, coronavirus

## Abstract

Coronavirus disease 2019 (COVID-19) is associated with high risk of malnutrition, primarily in older people; assessing nutritional risk using appropriate screening tools is critical. This systematic review identified applicable tools and assessed their measurement properties. Literature was searched in the MEDLINE, Embase, and LILACS databases. Four studies conducted in China met the eligibility criteria. Sample sizes ranged from six to 182, and participants’ ages from 65 to 87 years. Seven nutritional screening and assessment tools were used: the Nutritional Risk Screening 2002 (NRS-2002), the Mini Nutritional Assessment (MNA), the MNA-short form (MNA-sf), the Malnutrition Universal Screening Tool (MUST), the Nutritional Risk Index (NRI), the Geriatric NRI (GNRI), and modified Nutrition Risk in the Critically ill (mNUTRIC) score. Nutritional risk was identified in 27.5% to 100% of participants. The NRS-2002, MNA, MNA-sf, NRI, and MUST demonstrated high sensitivity; the MUST had better specificity. The MNA and MUST demonstrated better criterion validity. The MNA-sf demonstrated better predictive validity for poor appetite and weight loss; the NRS-2002 demonstrated better predictive validity for prolonged hospitalization. mNUTRIC score demonstrated good predictive validity for hospital mortality. Most instruments demonstrate high sensitivity for identifying nutritional risk, but none are acknowledged as the best for nutritional screening in older adults with COVID-19.

## 1. Introduction

Disease caused by the novel coronavirus 2019 (COVID-19) is currently the most serious public health issue on several continents [1]. Older age (over 65 years) has been associated with more severe disease and worse prognoses [2,3]. Factors such as the presence of comorbidities, greater propensity for systemic organ dysfunction, and poor nutritional status can contribute to the heightened risk of clinical complications in older adults with COVID-19 [4].

The course of COVID-19 presents with one severe inflammatory condition characterized by the involvement of proinflammatory cytokines [5]. Malnutrition may compromise the patient’s immune system and, consequently, the effectiveness of treatment, resulting in greater disease severity [6,7]. In this sense, the early assessment of nutritional risk, defined as ‘chances of a better or worse outcome from disease or surgery according to actual or potential nutritional and metabolic status’ [8], can contribute to the determination of the most appropriate nutritional therapy. Thus, adequate nutrition can provide a better immune system response and a more favorable prognosis [6,7].

There is no gold standard for identifying nutritional risk or malnutrition. In most cases, nutritional risk is researched via screening tools, typically applied by nutritionists, doctors, or other professionals, preceding a full nutritional assessment [9,10]. Tools such as the Mini Nutritional Assessment-short form (MNA-sf) [11,12], the Geriatric Nutrition Risk Index (GNRI) [13], the Nutritional Risk Screening 2002 (NRS-2002) [14], the Malnutrition Universal Screening Tool (MUST) [15], the Nutritional Risk Index (NRI) [16], the Short Nutritional Assessment Questionnaire (SNAQ) [17], and Nutrition Risk in the Critically Ill (NUTRIC) scores [18] are practical and inexpensive to apply and can predict clinical outcomes [9]. Nutritional assessment instruments, such as the Global Assessment Subjective (SGA) and the Mini Nutritional Assessment (MNA), also assess biochemical and laboratory parameters and clinical and dietetical factors [9,10]. The instruments MNA [19,20], MNA-sf [11,12,21], GNRI [13,21], MUST [15,21], and NRI [16,21] were developed specifically to identify nutritional risk or malnutrition in older adults, and others are also used to evaluate this target population [22,23].

Identifying older adults at nutritional risk proves difficult in the intensive care unit (ICU) due to the nature of critical illness. No consensus exists on the most appropriate method to identify these patients. Traditional screening and assessment tools did not uniformly identify older adults as malnourished or at nutritional risk in the ICU and, therefore, may be inappropriate for use in this population [24].

Several studies have recorded a high prevalence of nutritional risk and malnutrition in the population over 65 years old, especially those who are hospitalized [25,26]. Researchers have thus noted the need to assess nutritional risk among older adults with COVID-19 [27]. However, to our knowledge, no systematic review has yet been published of the nutritional screening instruments that could be used to identify nutritional risk in older adults with COVID-19. Therefore, the present study aims to identify the screening instruments that can be used to assess nutritional risk in older adults with COVID-19, and to clarify their measurement properties.

## 2. Materials and Methods

This systematic review included studies published in any language between November 2019 and July 2020, and was prepared following the recommendations of the preferred reporting items for systematic reviews and meta-analyses (PRISMA) [28]. The systematic review protocol is registered in PROSPERO (CRD42020186866).

### 2.1. Eligibility Criteria

Observational studies (cross-sectional, longitudinal, case series, case–control), clinical trials, comparative studies of different nutritional risk screening instruments, or validation studies that used nutritional screening instruments to identify nutritional risk in people over 65 years of age with COVID-19 were considered eligible for inclusion in the study. Review articles were excluded.

### 2.2. Databases and Search Strategy

The literature search was conducted on July 3, 2020, using the Medical Literature Analysis and Retrieval System Online database (MEDLINE, via PubMed), the Excerpta Medica Database (Embase), and the Latin American and Caribbean Health Sciences Literature (LILACS) database using a combination of the following descriptors and their synonyms: “nutrition risk”, “nutritional risk”, “nutritional screening”, “nutrition screening”, “nutritional assessment”, “nutritional index”, “Geriatric Nutrition Risk Index”, “Mini Nutritional Assessment”, “Subjective Global Assessment”, “Nutritional Risk Screening tool 2002”, “Malnutrition Universal Screening Tool”, “Nutritional Risk Index”, “Short Nutritional Assessment Questionnaire”, “Nutrition Risk in Critically Ill score”, and “COVID-19”. Appendix A presents the search strategy used in each database.

### 2.3. Screening and Selection of Studies

Two researchers independently (DS and SL) screened the studies. Initially, titles and abstracts were read. The full text of selected articles was then analyzed to confirm fulfillment of the study’s eligibility criteria.

### 2.4. Data Extraction

The following data were extracted from the selected articles: authorship; country and period of research; type of study; number, age, and sex of participants; criteria used to diagnose COVID-19; nutritional screening instrument used; properties of the nutritional screening instrument used (sensitivity, specificity, criterion validity, predictive validity); and participants’ nutritional risk information.

### 2.5. Evaluation of Studies’ Methodological Quality and Instruments’ Properties

The Newcastle–Ottawa scale was used to assess the methodological quality of retrospective studies [29], applying the modified version [30] to assess cross-sectional studies. The scale consists of eight questions; all are rated up to one star each, except for the question about comparability, which can be rated up to two stars. The modified version for cross-sectional studies consists of seven questions rated up to one star each, except those on questions on comparability and outcome, which can be rated up to two stars. Total scores for both versions can reach a maximum of nine stars. To assess case series studies, the instrument proposed by Murad et al. [31] was employed.

The nutritional screening instruments’ properties were evaluated according to the Quality Assessment of Diagnostic Accuracy Studies (QUADAS) criteria [32], which assess four domains: patient selection, instrument testing, reference standard and flow, and time of administration. The answers to the questions in each domain can be used to identify instrument’s the risk of bias and applicability [32].

### 2.6. Narrative Summary of Results

Descriptive analysis was performed to synthesize the data on the instruments’ properties (criterion validity, construct validity, and predictive validity) analyzed in each study. One narrative synthesis comparing the same properties between the different nutritional risk screening instruments was conducted. The prevalence of nutritional risk and malnutrition was compared between studies, considering the instruments used in each one.

The instruments’ sensitivity, specificity, positive predictive value (PPV), and negative predictive value (NPV) were evaluated. Sensitivity refers to the instrument’s ability to accurately identify individuals at nutritional risk (true positive [TP]/true positive [TP] + false negative [FN]) [33]. Specificity refers to the instrument’s ability to accurately identify well-nourished individuals (without malnutrition) as having no nutritional risk (true negative [TN]/true negative [TN] + false positive [FP]) [33].

PPV indicates the proportion of subjects who tested positive for nutritional risk and were actually at risk (true positive results; TP/TP + FP). NPV indicates to the proportion of subjects who tested negative for nutritional risk and were actually without risk (true negative results; TN/TN + FN) [34,35]. Sensitivity, specificity, PPV, and NPV were calculated based on the studies’ true positive, true negative, false positive, and false negative results. Statistical analysis, including calculation of 95% confidence intervals (CIs), was performed using RevMan (Review Manager) version 5.3. Sensitivity and specificity levels of over 80% were classified as good; levels above 50% and up to 80% were classified as weak; levels below 50% were classified as poor [36].

Criterion or construct validity of the nutritional screening instrument was analyzed for studies that compared the nutritional screening results with a reference criterion, according to the parameter or criterion used for comparison. Because no gold standard has been defined for diagnosis of nutritional risk [25,37], the methods or criteria suggested by Van Bokhorst-de van der Schueren et al. [36] were considered appropriate for analysis of criterion validity: nutritional assessment and anthropometry, objective assessment by a professional, or comparison with another nutritional screening tool of reference (e.g., the MNA or SGA).

Construct validity was analyzed for studies that compared the nutritional screening results with reference standards considered to have less validity, which have differences in comparison with nutritional screening instruments [9,36,38]. Evaluation was conducted of the relationship between the problem diagnosis (nutritional risk) and related variables not included in the instrument, for example, by analyzing in comparison to other nutritional screening instruments, such as the MNA-sf, MUST, NRS-2002, NRI, or modified NUTRIC (mNUTRIC) score, or the results of laboratory tests, including albumin, pre-albumin, creatinine, and total lymphocyte count (TLC) [39].

Predictive validity, referring to the screening tool’s ability to predict clinical outcomes, was assessed according to cutoff points of the area under the curve (AUC). When analyzing predictive validity, higher AUC results indicate greater ability to predict an outcome. In this study, the discriminative power of AUC was defined according to the traditional academic point system: 90 to 100 (excellent), 80 to 90 (good), 70 to 80 (weak), 60 to 70 (poor), and 60 or below (failure) [40]. This classification has been used in previous studies validating nutritional screening instruments [41,42].

Considering the number and heterogeneity of the studies included in this review, meta-analysis of the prevalence of nutritional risk or malnutrition was not possible.

## 3. Results

The initial database search retrieved 101 records; 13 duplicate studies were removed. Of the 88 remaining articles, which were analyzed by reading the title and abstract, 5 were selected for full-text review. Of these, four studies [43,44,45,46], all conducted in China, met the criteria for inclusion in the study: two retrospective cohort studies [43,46], a cross-sectional study [44], and a case series study [45]. Figure 1 shows the study selection flowchart.

### 3.1. Methodological Quality of Studies

The studies evaluated according to the Newcastle–Ottawa scale showed good methodological quality, with total scores of seven stars. The case series study also showed good methodological quality. Of the five questions applicable to this study, four were scored. Table 1 presents the assessment of methodological quality according to study type.

The studies’ quality with respect to the nutritional screening instruments employed was evaluated according to the QUADAS criteria. With respect to selection of participants, three studies [44,45,46] were at high risk of bias, as they did not perform random or consecutive sampling; one was at low risk of bias [43]. Additionally, the study by Yuan et al. [45] did not report data on the nutritional risk of two patients.

Low risk of bias was observed in all the studies with respect to evaluation of the nutritional screening results (index test domain) [43,44,45,46], as they defined a cut-off point for the diagnosis of nutritional risk. Two studies compared between the diagnosis of nutritional risk and patients’ body mass index (BMI) [44,45,46], but it was clear that this did not interfere in the diagnosis of patients’ nutritional risk.

Two studies [44,45,46] compared diagnoses of nutritional risk to other nutritional screening tools as well as BMI. In these studies, a low risk of bias was identified with respect to the reference standards used and for the sequence and timing of the comparisons. Although BMI may not be one of the best indicators of nutritional risk when used alone, the study authors adequately addressed the limitations of using BMI to identify nutritional risk or malnutrition, as well as the limitations regarding BMI’s predictive validity. Table 2 presents the evaluation of the studies’ quality with respect to the instruments used.

### 3.2. Participants’ Characteristics

The number of participants in each study ranged from six [45] to 182 [44], and ages ranged from 65 [44,46] to 87 years [46]. Three studies included more women than men [44,45,46]. Table 3 illustrates the characteristics of the participants and instruments used in each study.

### 3.3. COVID-19 Diagnosis Method

Li et al. [44] diagnosed COVID-19 according to a positive reverse-transcriptase polymerase chain reaction (RT-PCR) test for SARS-CoV-2. Liu et al. [46] diagnosed and classified COVID-19 using a combination of several criteria, including history of epidemiological exposure; characteristic symptoms of COVID-19, such as fever, cough, and gastrointestinal symptoms; laboratory tests with changes in lymphocyte count and number of white blood cells; changes in pulmonary imaging tests; and a positive RT-PCR test result for SARS-CoV-2 using respiratory or blood specimens.

Yuan et al. [45] used only RT-PCR test results to diagnose COVID-19 in two patients; PCR-RT test results and history of epidemiological exposure in two other patients; and a combination of clinical, epidemiological, and laboratory test information for the final two patients.

Zhang et al. [43] diagnosed and clinically classified COVID-19 according to the *Guidance for Coronavirus Disease 2019* (6th edition) published by the National Health Commission of China, which includes criteria related to the RT-PCR test and the characteristic symptoms of COVID-19.

### 3.4. Nutritional Screening Instruments Used to Identify Nutritional Risk

The studies used various nutritional screening tools to identify nutritional risk in older people with COVID-19. Liu et al. [46] used four different instruments: the NRS-2002, MNA-sf, MUST, and NRI. Cut-off scores for diagnosis of nutritional risk were ≥3 points for the NRS-2002 (out of 6) [14], <12 for the MNA-sf (out of 14) [11,12], ≥2 for the MUST (out of 6) [15], and <83.5 for the NRI (severe risk of malnutrition; a score of >100 indicates no risk) [16].

Li et al. [44] used the full version of the MNA to assess the nutritional risk, with three classifications: no risk of malnutrition (≥24 points), risk of malnutrition (17–23.5 points), and malnutrition (<17 points).Yuan et al. [45] used the GNRI, with four categories: high risk, the cut-off point used to diagnose nutritional risk (<82 points), moderate risk (82–91 points), low risk (92–98 points), and no risk (>98 points). Zhang et al. [43] used mNUTRIC scores with two categories: high nutritional risk (≥5 points) and low nutritional risk (<5 points). Table 4 presents the parameters for each instrument.

### 3.5. Nutritional Risk in Older Adults with COVID-19

Based on the various instruments applied, the prevalence rates of nutritional risk in Liu et al.’s study [46] were 85.1% (NRS-2002), 77.3% (MNA-sf), 60.4% (NRI), and 41.1% (MUST) [46]. Li et al. found that 50 patients (27.5%) were at risk of malnutrition and 96 (52.7%) were malnourished [44]. Based on mNUTRIC scores, Zhang et al. found that 61% of patients had a high nutritional risk [43]. All four participants with reported GNRI evaluation data were classified by Yuan et al. as at high nutritional risk [45].

### 3.6. Association between Comorbidities and Nutritional Risk

Li et al. found that diabetes mellitus was associated with higher nutritional risk and malnutrition (odds ratio (OR): 2.12, 95% CI: 1.92–3.21) [44]. Liu et al. found no significant associations between arterial hypertension, vascular and cerebrovascular diseases, and nutritional risk [46]. Zhang et al. found no significant associations between nutritional risk and hypertension, diabetes, cardiovascular disease, malignancy, chronic obstructive pulmonary disease, chronic kidney disease, liver cirrhosis, or immunopathy [43].

### 3.7. Sensitivity, Specificity, and Criterion Validity of Nutritional Screening Instruments

Liu et al. [46] compared diagnoses of nutritional risk between the different instruments and with cut-off points of BMI. Compared to the cut-off points for BMI, all the instruments (NRS-2002, MNA-sf, MUST, and NRI) demonstrated 100% sensitivity. Between the instruments, only the MUST showed a sensitivity of less than 50% compared to the NRS-2002.

Only the MUST showed better specificity compared to BMI (62%). Between the instruments, good specificity was identified for the MUST vs. the NRS-2002 (95%), the MUST vs. the NRI (88%), the MNA-sf vs. the NRS-2002 (86%), and the MNA-sf vs. the MUST (97%).

The NRS-2002 was the only instrument to present an NPV above 80% compared to BMI, the MUST, and the NRI. All instruments had a PPV below 13% when compared to BMI. The GNRI demonstrated a PPV of 25%.

The MNA showed weak criterion validity, demonstrating a significant correlation with BMI, calf circumference, albumin, and TLC but no correlation with tricipital skinfold thickness and arm circumference. The MUST, NRS-2002, MNA-sf, and NRI all showed poor criterion validity compared to BMI. The GNRI demonstrated poor construct validity when compared to TLC. mNUTRIC score demonstrated poor construct validity, having a significant correlation with TLC and creatinine but no correlation with albumin and pre-albumin levels. Table 5 reports the diagnostic performance of the nutritional screening and assessment instruments included in this review.

### 3.8. Predictive Validity of Screening and Nutritional Assessment Instruments

In the study by Liu et al. [46], patients with nutritional risk according to the NRS-2002, MNA-sf, and NRI exhibited longer hospitalization, worse disease, worse appetite, and greater weight change than patients without nutritional risk. Diagnosis of nutritional risk according to the MUST predicted worse appetite and weight change, but not of worsening disease or prolonged hospitalization.

Zhang et al. [43] found that mNUTRIC score predicted complications in the ICU such as acute respiratory distress syndrome (ARDS), shock, acute myocardial injury, secondary infection, and death in the ICU after 28 days of hospitalization, but did not predict acute liver dysfunction, embolization/thrombosis, acute kidney injury, or pneumothorax.

The NRS-2002 had poor predictive validity for patients’ length of hospital stay and poor validity for hospital expenses, decreased appetite, and weight loss of more than 2.6 kg. The MNA-sf had good predictive validity for changes in appetite and weight, but weak predictive validity for length of hospital stay, and failure for hospital expenses. The MUST had good predictive validity for weight change and poor predictive validity for appetite change. The NRI demonstrated poor predictive validity for all analyzed outcomes. mNUTRIC scores showed good predictive validity for complications in the ICU such as ARDS, shock, acute myocardial injury, secondary infection, and ICU mortality after 28 days of hospitalization. Table 6 presents the analysis of the predictive validity of the nutritional screening instruments included in this review.

## 4. Discussion

To our knowledge, this is the first systematic review to verify the nutritional screening tools used to identify nutritional risk in older adults with COVID-19. The instruments evaluated in this review can be considered useful for assessing nutritional risk in older adults with COVID-19. Although the majority (except the MUST) demonstrated low specificity, they showed high sensitivity and/or good predictive validity. The NRS-2002 was the only instrument to demonstrate a sensitivity of 100% compared to BMI and greater than 90% compared to three other nutritional screening instruments (the MNA-sf, MUST, and NRI). However, the MUST demonstrated better specificity. The MNA and the MUST showed better criterion validity. When tools with high sensitivity and specificity are not available, high sensitivity to identify risk is the preferable criterion for selecting a screening instrument because this is essential for directing individualized nutritional management with a focus on improving the patient’s nutritional status [47,48].

Different nutritional screening instruments identified different prevalence rates of nutritional risk. Liu et al. [46] found a prevalence of 85.1% using the NRS-2002, but only 41.1% using the MUST. Previous studies have also reported overestimation by the NRS-2002 in identifying nutritional risk in older individuals [21,49]. For example, Poulia et al. [21] found that the NRS-2002 showed greater sensitivity (99%) and less specificity (6.1%) when identifying nutritional risk in older adults compared to the NRI, GNRI, MNA-sf, MUST, and SGA.

The evaluation criteria of the NRS-2002 and MUST can explain the wide difference in the prevalence of nutritional risk they report in older adults with COVID-19. The Multinational Consensus Statement from the Fleischner Society reports that patients with moderate and severe clinical conditions can present significant lung dysfunction or damage [50]. Chen et al. [51] have reported that 43.6% of older adults with COVID-19 had severe conditions at hospital admission, and 43.6% were critical cases. Likewise, in a study that evaluated 24 COVID-19 patients in critical condition, all were admitted to the ICU with hypoxemic respiratory failure and with 75% requiring mechanical ventilation [52]. The severe respiratory condition of most patients hospitalized with COVID-19 may favor higher scores in terms of disease severity (2 or 3 points). This, in turn, can affect the NRS-2002′s identification of nutritional risk in these patients. This can be seen in a study by Zhao et al. [53], which included 371 young adults and older adults with COVID-19 in severe and critical condition. Disease severity was scored as 2 points for 99% of the patients in severe condition (307 of 310) and 3 points (the maximum) for 100% (61) of the patients in critical condition.

In the nutritional score domain of the NRS-2002, patients receive 1 point if they demonstrate weight loss greater than 5% in the last three months, or food intake between 50% and 75% of nutritional needs (from a maximum score of 3). These conditions are common among older adults with COVID-19, which may explain the impact of this domain in characterizing nutritional risk (score ≥ 3) according to the NRS-2002 [14]. For older adults over 70, an additional point is awarded for weight loss of more than 5% in the last three months or food intake between 50% and 75% of nutritional needs, leading to a diagnosis of nutritional risk [14].

The MUST, in contrast, has three domains: BMI, weight loss, and consequences of disease severity. Inclusion of BMI as a parameter for classifying nutritional risk may represent limited applicability of this nutritional screening instrument for hospitalized older adults; it is difficult to measure weight and height, needed to calculate BMI, in bedridden patients [9,54]. Furthermore, there are still limitations in the accuracy and precision of the weight and height estimation equations used for nutritional assessment among older adults [55].

The MUST is assigned a point in the BMI domain for people with a BMI less than 20 kg/m^2^. Several studies have reported a high prevalence of overweight and obesity among patients with COVID-19, with few older adults with COVID-19 having low weight [56,57]. In a study including 37 patients with a mean age over 60 years, 23 were overweight (BMI > 24.9 kg/m^2^), 13 were adequate weight (BMI between 18.5 and 24.9 kg/m^2^), and only 1 was underweight (BMI < 18.5 kg/m^2^) [57]. Because the MUST is applied on admission and weight loss is more visible during hospitalization, it could be that few older adults with COVID-19 meet this criterion [58]. Furthermore, the MUST’s low BMI cut-off point may not be adequate to identify nutritional risk related to weight loss in older adults, because the optimal BMI is higher in this population than the optimum proposed for young adults [9].

In the weight loss domain, points are only awarded if weight loss is between 5% and 10% in the last three to six months. This criterion can also be difficult to identify in some people with COVID-19, considering that many times, the people does not know how to provide this information, or the records are not available on the patient’s admission. Furthermore, COVID-19 can worsen very quickly in older adults [59]. Wang et al. [59] found that older Chinese with COVID-19 showed rapid disease progression, with an average duration of 11.5 days from first symptoms to death. Thus, weight loss above 5% may not occur before hospital admission.

The disease severity domain is assigned 2 points for people with a drastic reduction of food consumption or inability to eat on more than five days. Chen et al. [51] found that 9.1% of older adults with COVID-19 experienced anorexia. Thus, this area could be assigned a low score before these symptoms are present and severe enough to fulfill the criterion.

mNUTRIC scores showed poor construct validity when compared to albumin, pre-albumin, creatinine, and TLC levels. Despite disagreement among researchers on the use of these laboratory parameters to assess the validity of nutritional screening instruments, some studies have reported them to associate with the diagnosis of nutritional risk [60,61], and other systematic reviews have used them for the analysis of construct validity [9,36,38]. These parameters are also included in some instruments, such as CONUT [62], the Nutrition Screening Equation [63], the GNRI [13], and the NRI [16].

These laboratory parameters are influenced by several factors, including inflammation and acute disease [64,65,66,67]. Zhang et al. [43] recorded low concentrations of albumin, pre-albumin, and TLC in people with both high and low nutritional risk. This may have been because the mNUTRIC score is an instrument for critically ill patients, and a severe inflammatory condition is recorded for people with COVID-19 in this context [68,69].

Many authors question the validity of albumin as an indicator of nutritional status because it has a half-life of 20 days [70,71], whereas other researchers consider this half-life applicable under normal physiological conditions. In cases of disease, including infection, the half-life of albumin can be reduced [72,73]. Thus, analysis of albumin concentrations requires caution, especially in people with COVID-19 who have been administered corticosteroids, which can further modify albumin concentrations [74,75].

The prevalence of nutritional risk was high in all studies, ranging from 27.5% [44] to 100% [45]. Only one study evaluated actual malnutrition, identifying a prevalence of 52.7% [44]. These prevalence rates are higher than those recorded in previous literature regarding hospitalized older adults [20,23], including Chinese adults [76]. A previous systematic review identified a 22.0% prevalence of malnutrition in hospitalized older adults based on the MNA [20]. In contrast, Li et al. [44], who also used the MNA, identified a 52.7% prevalence of malnutrition in older adults with COVID-19, in addition to 27.5% prevalence of nutritional risk. Another previous study involving 339 older people in Nepal identified a 49.6% prevalence of nutritional risk and a 24.8% prevalence of malnutrition [77].

Using the NRS-2002, Liu et al. [46] identified an 85.1% prevalence of nutritional risk among older adults with COVID-19. This is higher than the prevalence recorded by Zhang et al. [78], who also used the NRS-2002, but in a study that included 536 hospitalized patients with different diseases. Another study using the NRS-2002, included 114 Chinese with gastric cancer, a clinical condition that leads to severe catabolism, identified a 70.1% prevalence of nutritional risk [76], and a systematic review including studies of European patients hospitalized for various diseases found a prevalence of 41.5% [23].

Li et al. [44], who evaluated the association between comorbidities and nutritional risk [43,46], found an association between diabetes mellitus and nutritional risk. No significant associations were identified with other comorbidities analyzed in the studies. Diabetes mellitus is associated with greater nutritional risk in the older population in general [79]. Paris et al. [80] identified a 39.1% prevalence of nutritional risk in older hospitalized patients with diabetes, and a 21.2% prevalence of malnutrition. Diabetes mellitus is also a risk factor for COVID-19 progression and worse prognosis [81,82]. Thus, particular focus should be directed toward the nutritional screening process of hospitalized older adults with COVID-19 with comorbid diabetes mellitus, many of whom may also be obese, another risk factor for worsening the disease [83,84]. In a systematic review, 38 of the 44 studies reported a prevalence of obesity above 38% in hospitalized patients with an average age between 50 and 70 years and diabetes mellitus [85]. Considering that both diabetes and nutritional risk relate to worsening progression of COVID-19, it is important that personalized nutritional therapy for older adults with COVID-19 considers all these factors [7,27,59].

Accordingly, considering BMI alone or as the primary criterion in nutritional screening may neglect or underestimate nutritional risk among people with COVID-19 with a high BMI [86]. In the study by Liu et al. [46], for example, overweight people were diagnosed with nutritional risk according to the NRS-2002, MUST, MNA- sf, and NRI. This highlights the importance of using nutritional screening tools to identify nutritional risk in older adults with COVID-19, regardless of BMI. 

Liu et al. [46] found that a diagnosis of nutritional risk using the NRS-2002, NRI, and MNA-sf predicted longer hospital stays, worse disease, worse appetite, and greater weight change as compared to no nutritional risk. mNUTRIC score predicted complications in the ICU such as ARDS, shock, acute myocardial injury, secondary infection, and mortality after 28 days of hospitalization. The good predictive validity of the mNUTRIC score for ICU complications can be explained by the fact that this instrument was developed to screen for nutritional risk in critically ill patients, with a focus on identifying clinical complications related to nutritional risk [18,87]. Thus, the good predictive validity of mNUTRIC scores can contribute to better monitoring of patients’ prognosis and more effective treatment by the health team.

The predictive validity of the screening instruments was weak or poor for other factors, except for the MNA-sf, which showed good predictive validity for poor appetite and weight change, and the mNUTRIC score, which showed good predictive validity for mortality in the ICU after 28 days of hospitalization. A diagnosis of nutritional risk according to the MUST predicted only poor appetite and weight change. This may be related to the evaluation criteria of this instrument, which include changes in body weight and appetite (reduction of food consumption) as important factors. Several previous studies have also reported on the predictive validity of nutritional screening instruments for various clinical outcomes [88,89,90,91].

In addition to assessing older people’ nutritional risk upon hospital admission and during treatment of COVID-19, follow-up after improvement of or recovery from the disease with remission of pneumonia is necessary. A study including 50 young and older adult patients in a hospital unit and in rehabilitation after recovery from COVID-19 identified that 45% were at high nutritional risk and 26% were at moderate risk, according to MUST criteria [92].

During the COVID-19 pandemic, with the adoption of physical distancing measures, situations may arise that require new approaches to nutritional screening of older adults who have recovered from the disease. For countries in which telemedicine is authorized, nutritional screening can be conducted at a distance. To this end, Krznarić et al. [93] proposed a simple and practical protocol for assessing malnutrition in adult and older adults. The Remote Malnutrition App (R-MAPP) includes assessment of nutritional risk with MUST adaptation and can be used in primary health care settings.

This systematic review presents the strengths and weaknesses of various instruments used to assess nutritional risk in older adults with COVID-19. However, some limitations must be considered. First, all the included studies were conducted in China. Caution is required when extrapolating the results to other populations. Second, the sample sizes in the four studies were small, and selection was not random or consecutive. Third, there was no standardization among the studies regarding the gold standard diagnostic criterion for COVID-19 (RT-PCR test). Finally, because the studies were not prospective and controlled, confusion and selection biases may have influenced the results, and there is no way to infer causality regarding the diagnosis of COVID-19 and nutritional risk or malnutrition.

## 5. Conclusions

Nutritional risk was highly prevalent among older adults with COVID-19 regardless of the nutritional screening tool applied. The NRS-2002, MNA, MNA-sf, NRI, and MUST showed high sensitivity, but only the MUST demonstrated better specificity compared to BMI. Of the various instruments used in the studies, the MNA and MUST presented the best criterion validity. The MNA-sf had the best predictive validity for poor appetite and weight loss over 2.6 kg, and the NRS-2002 had the best predictive validity for the length of hospital stay. mNUTRIC score had good predictive validity for complications in the ICU such as ARDS, shock, acute myocardial injury, secondary infection, and mortality after 28 days of hospitalization.

Considering their convenience, low cost, and good ability to predict clinical outcomes, nutritional screening and assessment tools can contribute to the early diagnosis of people with greater nutritional risk. Because nutritional risk is a modifiable factor that can be reduced or controlled with early, individualized nutritional therapy, identifying risk using instruments with adequate sensitivity can help prevent worsening disease and improve patients’ prognoses.

## Figures and Tables

**Figure 1 nutrients-12-02956-f001:**
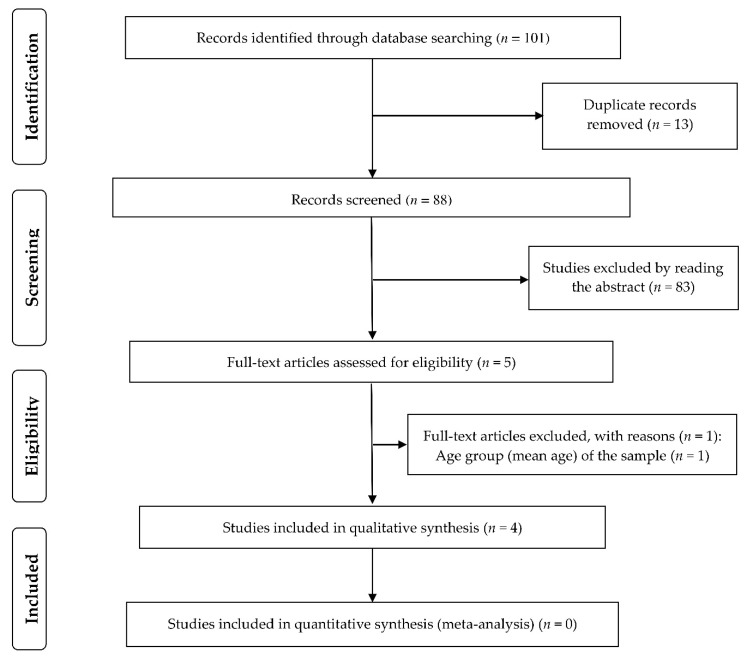
The preferred reporting items for systematic reviews and meta-analyses (PRISMA) flow chart for selection of studies.

**Table 1 nutrients-12-02956-t001:** Methodological quality of included studies (*n* = 4).

Cohort—Newcastle-Ottawa Scale									
	Selection				Comparability		Outcome		
Quality criteria	1. Representativeness of the exposed cohort	2. Selection of the non-exposed cohort	3. Ascertainment of exposure	4. Demonstration that outcome of interest was not present at start of study	1. Comparability of cohorts on the basis of the design or analysis		1. Assessment of outcome	2. Was follow-up long enough for outcomes to occur?	3. Adequacy of follow up of cohorts
Liu et al., 2020 [46]	*		*		**		*	*	*
Zhang et al., 2020 [43]	*		*		**		*	*	*
Cross-sectional—Newcastle–Ottawa Scale									
	Selection				Comparability		Outcome		
Quality criteria	1. Representativeness of the sample	2. Sample size	3. Ascertainment of exposure	4. Non-respondents	1. The subjects in different outcome groups are comparable, based on the study design or analysis. Confounding factors are controlled.		1. Assessment of outcome		2. Statistical test
Li et al., 2020 [44]	*		*		**		**		*
Case series—Murad et al. (2018)									
	Selection	Ascertainment		Causality					Reporting
Quality criteria	1. Does the patient (s) represent (s) the whole experience of the investigator (center), or is the selection method unclear to the extent that other patients with similar presentation may not have been reported?	2. Was the exposure adequately ascertained?	3. Was the outcome adequately ascertained?	4. Were other alternative causes that may explain the observation ruled out?	5. Was there a challenge/rechallenge phenomenon?	6. Was there a dose–response effect?	7. Was follow-up long enough for outcomes to occur?		8. Is the case (s) described with sufficient details to allow other investigators to replicate the research or to allow practitioners make inferences related to their own practice?
Yuan et al., 2020 [45]		*	*	NA	NA	NA	*		*

Notes: for case series, items 4, 5 and 6 are mostly relevant to cases of adverse drug events; *, one star; **, two stars. Abbreviations: NA, Not Applicable.

**Table 2 nutrients-12-02956-t002:** Methodological quality evaluation of each study included in the systematic review according to Quality Assessment of Diagnostic Accuracy Studies (QUADAS; *n* = 4).

Domain		Item	Liu et al., 2020 [46]	Li et al., 2020 [44]	Yuan et al., 2020 [45]	Zhang et al., 2020 [43]
Patient Selection	Signaling questions (yes/no/unclear)	Was a consecutive or random sample of patients enrolled?	No	No	No	Yes
		Was a case–control design avoided?	Yes	Yes	Yes	Yes
		Did the study avoid inappropriate exclusions?	Yes	Yes	No	Yes
	Risk of bias: High/low/unclear	Could the selection of patients have introduced bias?	High	High	High	Low
	Concerns regarding applicability: High/low/unclear	Are there concerns that the included patients do not match the review question?	Low	Low	Low	Low
Index Test	Signaling questions (yes/no/unclear)	Were the index test results interpreted without knowledge of the results of the reference standard?	Unclear	-	-	-
		If a threshold was used, was it pre-specified?	Yes	Yes	Yes	Yes
	Risk of bias: High/low/unclear	Could the conduct or interpretation of the index test have introduced bias?	Low	Low	Low	Low
	Concerns regarding applicability: High/low/unclear	Are there concerns that the index test, its conduct, or interpretation differ from the review question?	Low	Low	Low	Low
Reference Standard	Signaling questions (yes/no/unclear)	Is the reference standard likely to correctly classify the target condition?	Yes	Yes	-	-
		Were the reference standard results interpreted without knowledge of the results of the index test?	Unclear	Unclear	-	-
	Risk of bias: High/low/unclear	Could the reference standard, its conduct, or its interpretation have introduced bias?	Low	Low	-	-
	Concerns regarding applicability: High/low/unclear	Are there concerns that the target condition as defined by the reference standard does not match the review question?	Low	Low	-	-
Flow and Timing	Signaling questions (yes/no/unclear)	Was there an appropriate interval between index test (s) and reference standard?	Yes	Yes	-	-
		Did all patients receive a reference standard?	Yes	Yes	-	-
		Did all patients receive the same reference standard?	Yes	Yes	-	-
		Were all patients included in the analysis?	Yes	Yes	-	-
	Risk of bias: High/low/unclear	Could the patient flow have introduced bias?	Low	Low	-	-

**Table 3 nutrients-12-02956-t003:** Characteristics of participants included in the studies (*n* = 4).

Author	Country	Design	*n*	Age Group (Years)	Sex	Nutritional Screening Tool	Nutritional Risk
Liu et al., 2020 [46]	China	Retrospective cohort	141	65 to 87	Women: 73. Men: 68	NRS-2002-NR: score ≥ 3 (out of a maximum of 6); MNA-sf-NR: score < 12 (out of a maximum of 14); MUST-NR: score ≥ 2 (out of a maximum of 6); NRI-SNR: score < 83.5 and no NR > 100.	NRS-2002: 120 (85.1%); MNA-sf: 109 (77.3%); MUST: 58 (41.1%); NRI: 101 (60.4%).
Li et al., 2020 [44]	China	Cross-sectional	182	Average age of 68.5 years old	Women: 117. Man: 65.	MNA: No NR/malnutrition ≥ 24; risk of malnutrition: 17–23.5; malnutrition < 17.	No nutritional risk/malnutrition: 36 (19.8%); risk of malnutrition: 50 (27.5%); malnutrition: 96 (52.7%).
Yuan et al., 2020 [45]	China	Case series	6^1^	65 to 71	Women: 4. Man: 2.	GNRI: High NR: score < 82—cut-off point used to diagnose nutritional risk in the study; moderate NR: score from 82 to <92; low NR: score from 92 to ≤98; no risk: score > 98.	4 (100%).
Zhang et al., 2020 [43]	China	Retrospective cohort	136	Average age 69 years	Women: 50 (37%) Man: 86 (63%)	mNUTRIC score. High NR ≥ 5. Low NR < 5.	High NR: 83 (61.0%). Low NR: 53 (39.0%).

Abbreviations: GNRI, Geriatric Nutritional Risk Index; MNA, Mini Nutritional Assessment; MNA-sf, Mini Nutritional Assessment-short form; MUST, Malnutrition Universal Screening Tool; NRI, Nutritional Risk Index; NRS-2002, Nutritional Risk Screening tool 2002; NUTRIC score, modified Nutrition Risk in the Critically ill (mNUTRIC) score; NR, nutritional risk; SNR, severe nutritional risk. ^1^ For four participants, data on nutritional risk were provided.

**Table 4 nutrients-12-02956-t004:** Parameters of nutritional screening tools used in the included studies.

Tool	Criteria	Score	Applications
NRI	NRI = (1.519 × serum albumin (g/L) + 41.7 × (present weight/usual weight)	No NR > 100. Mild risk: 97.5–100. Moderate risk: 83.5–97.5. High risk < 83.5.	Recommended settings: hospital, and home care.
GNRI	GNRI = (14.89 × albumin (g/dL)) + (41.7 × (body weight/ideal body weight))	Low NR 92–≤98. Moderate NR: 82–<92. High NR <82.	Recommended settings: hospital.
MUST	Three domains: BMI, weight loss, and consequences of disease severity. Each parameter can be rated as 0, 1, or 2. BMI domain: BMI (kg/m^2^) > 20 (0), 18.5–20.0 (1), <18.5 (2). Unintentional weight loss in past 3–6 months (%): <5 (0), 5–10 (1), >10 (2). Disease severity domain: drastic reduction of food consumption or inability to eat on more than five days (2).	Low NR: 0. Medium NR: 1. High NR ≥ 2.	Recommended settings: hospital, home care, and community.
NRS-2002	Two domains: disease severity score and nutritional score. Disease severity score domain: patients with diabetes, cancer, COPD (1 point); patients with severe pneumonia (2 points); intensive care patients (APACHE > 10) (3 points). Nutritional score domain: Weight loss greater than 5% in the last three months or food intake between 50% and 75% of nutritional needs (1 point); weight loss greater than 5% in the last two months, food intake between 25% and 60% of nutritional needs, or BMI 18.5–20.5 with impaired general health (2 points); weight loss greater than 5% in one month, >15% in three months, or food intake between 0% and 25% of nutritional needs (3 points). Score adjusted for age: if ≥70 years, one additional point.	NR: score ≥ 3.	Recommended settings: hospital, home care, and community.
NUTRIC score	Six domains: age, APACHE, SOFA, number of comorbidities, days from hospital to ICU admission, and IL-6. Age: <50 (0); 50–74 (1); ≥75 (2). APACHE II: <15 (0); 15–19 (1); 20–28 (2); ≥28 (3). SOFA: <6 (0); 6–9 (1); ≥10 (2). Number of comorbidities: 0–1 (0); ≥2 (1). Days from hospital to ICU admission: 0–<1 (0); ≥(1). IL-6: 0-<400 (0); ≥400 (1).	Score with IL-6: Low NR: 0–5. High NR: 6–10. Score without IL-6: Low NR: 0–4. High NR: 5–9.	Recommended settings: critically ill patients (ICU).
MNA-sf	Six domains: decrease in food intake, weight loss, mobility, disease severity, neuropsychological problems (depression, dementia), and BMI. Decrease in food intake: severe (0); moderate (1); none (2). Involuntary weight loss during the last three months? >3 kg (0); does not know (1); 1–3 kg (2); none (3). Mobility: bedridden (0); able to get out of bed/chair but does not go out (1); goes out (2). Disease severity: acute disease or psychological stress in the past 3 months (0); no acute disease or psychological stress in the past 3 months (2). Neuropsychological problems: severe depression or dementia (0); mild dementia (1); none (2). BMI (kg/m^2^): <19 (0); <21 (1); <23 (2); ≥23 (3).	Normal: 12–14. Risk of malnutrition: 8–11. Malnutrition: 0–7.	Recommended settings: hospital, home care, and community.
MNA	18 domains: decrease in food intake, weight loss, mobility, disease severity, neuropsychological problems (depression, dementia), and BMI (For these domains, same criteria as in the MNA-sf.). Other domains: lives independently, medication, pressure sores or skin ulcers, number of full meals daily, markers for protein intake, fruit or vegetable consumption, fluid intake, mode of feeding, self-view of nutritional status, self-assessment of health status, mid-arm circumference in cm, and calf circumference in cm.	Normal: 24–30. At risk of malnutrition: 17–23.5. Malnutrition < 17.	Recommended settings: hospital, home care, and community.

Abbreviations: APACHE, Acute Physiology and Chronic Health Evaluation; BMI, body mass index; COPD, chronic obstructive pulmonary disease; GNRI, Geriatric Nutritional Risk Index; ICU, intensive care unit; IL-6, Interleukin-6; MNA, Mini Nutritional Assessment; MNA-sf, Mini Nutritional Assessment-short form; MUST, Malnutrition Universal Screening Tool; NRI, Nutritional Risk Index; NRS-2002, Nutritional Risk Screening tool 2002; NUTRIC score, modified Nutrition Risk in the Critically ill (mNUTRIC) score; NR, nutritional risk; SOFA, Sequential Organ Failure Assessment score.

**Table 5 nutrients-12-02956-t005:** Diagnostic performance of Nutritional Screening Tools identifying older adults with COVID-19 at nutritional risk or with malnutrition.

Author	Screening Tool	Reference Standard	TP	FP	FN	TN	Sensitivity (95%CI)	Specificity (95% CI)	PPV (%)	NPV (%)	Other Analysis
Liu et al., 2020 [46]	NRS-2002	BMI	7	113	0	21	100 (59 to 100)	16 (10 to 23)	5.8	100	-
Liu et al., 2020 [46]	NRS-2002	MUST	57	63	1	20	98 (91 to 100)	24(15 to 35)	47.5	95.2	-
Liu et al., 2020 [46]	NRS-2002	MNA-sf	102	18	7	14	94 (87 to 97)	44 (26 to 62)	85.0	66.7	-
Liu et al., 2020 [46]	NRS-2002	NRI	98	22	3	18	97 (92 to 99)	45 (29 to 62)	81.7	85.7	-
Liu et al., 2020 [46]	MUST	BMI	7	51	0	83	100 (59 to 100)	62 (53 to 70)	12.1	100	-
Liu et al., 2020 [46]	MUST	NRS-2002	57	1	63	20	47 (38 to 57)	95 (76 to 100)	98.3	24.1	-
Liu et al., 2020 [46]	MUST	MNA-sf	57	52	1	31	98 (91 to 100)	37 (27 to 49)	52.3	96.9	-
Liu et al., 2020 [46]	MUST	NRI	53	5	48	35	52 (42 to 63)	88 (73 to 96)	91.4	42.2	-
Liu et al., 2020 [46]	MNA-sf	BMI	7	102	0	32	100 (59 to 100)	24 (17 to 32)	6.4	100	-
Liu et al., 2020 [46]	MNA-sf	NRI	86	23	15	17	85 (77 to 91)	42 (27 to 59)	78.9	53.1	-
Liu et al., 2020 [46]	MNA-sf	NRS-2002	102	3	22	18	82 (74 to 99)	86 (64 to 97)	97.1	45.0	-
Liu et al., 2020 [46]	MNA-sf	MUST	57	1	52	31	52 (43 to 62)	97 (84 to 100)	98.3	37.4	-
Liu et al., 2020 [46]	NRI	BMI	7	94	0	40	100 (59 to 100)	30 (22 to 38)	6.9	100	-
Liu et al., 2020 [46]	NRI	MNA-sf	86	15	23	17	79 (70 to 86)	53 (35 to 71)	85.2	42.5	-
Liu et al., 2020 [46]	NRI	NRS-2002	98	3	22	18	82 (74 to 88)	86 (64 to 97)	97.0	45.0	-
Liu et al., 2020 [46]	NRI	MUST	53	48	35	5	60 (49 to 61)	9 (3 to 21)	52.5	12.5	-
Li et al., 2020 [44]	MNA	BMI	-	-	-	-	-	-	-	-	BMI (kg/m^2^)—no malnutrition: 25.6 ± 3.0; risk of malnutrition: 23.3 ± 3.4 kg/m^2^; malnutrition: 21.1 ± 3.6 kg/m^2^. F or X^2^ value: 4.106, *p* = 0.035.
Li et al., 2020 [44]	MNA	Calf circumference (cm)	-	-	-	-	-	-	-	-	Calf circumference (cm)—no malnutrition: 33.4 ± 5.6; risk of malnutrition: 31.2 ± 4.8; malnutrition: 28.7 ± 5.7, F or X^2^ value: 2.518, *p* = 0.047.
Li et al., 2020 [44]	MNA	Albumin (g/L)	-	-	-	-	-	-	-	-	Albumin (g/L)—no malnutrition: 38.5 ± 4.2; risk of malnutrition: 30.1 ± 6.4; malnutrition: 25.7 ± 5.3, F or X^2^ value: 10.217, *p* < 0.001.
Li et al., 2020 [44]	MNA	TLC	-	-	-	-	-	-	-	-	TLC—no malnutrition: 1.7 ± 0.52; risk of malnutrition: 1.2 ± 0.43, malnutrition: 0.9 ± 0.38, F or X^2^ value: 11.237, *p* < 0.001.
Li et al., 2020 [44]	MNA	TSFT (mm)	-	-	-	-	-	-	-	-	TSFT (mm)—no malnutrition: 16.8 ± 7.2; risk of malnutrition: 15.7 ± 6.9; malnutrition: 14.9 ± 7.3, F or X^2^ value: 1.632, *p* = 0.126.
Li et al., 2020 [44]	MNA	MAC (cm)	-	-	-	-	-	-	-	-	MAC (cm)—no malnutrition: 28.7 ± 2.8; risk of malnutrition: 27.6 ± 3.3; malnutrition: 26.5 ± 3.2, F or X^2^ value: 2.679, *p* = 0.379.
Yuan et al., 2020 [45]	GNRI	TLC	-	-	-	-	-	-	-	-	Of the four patients at nutritional risk, one had low TLC levels and three had normal levels.
Zhang et al., 2020 [43]	NUTRIC score	Albumin (g/L)	-	-	-	-	-	-	-	-	High NR group (*n* = 83): 29 g/L (25–32). Low NR group (*n* = 53): 30 g/L (28–32), *p* = 0.107.
Zhang et al., 2020 [43]	NUTRIC score	Prealbumin (g/L)	-	-	-	-	-	-	-	-	High NR group (*n* = 83): 82 g/L (80–122). Low NR group (*n* = 53): 95 (80–128) g/L, *p* = 0.281.
Zhang et al., 2020 [43]	NUTRIC score	TLC	-	-	-	-	-	-	-	-	High NR group (*n* = 83): 0.5 × 10^9^/L (0.3–0.7). Low NR group (*n* = 53): 0.6 × 10^9^/L (0.4–0.9), *p* = 0.007.
Zhang et al., 2020 [43]	NUTRIC score	Creatinine (mmol/L)	-	-	-	-	-	-	-	-	High NR group (*n* = 83): 90 (65–144) mmol/L. Low NR group (*n* = 53): 67 (54–85) mmol/L, *p* < 0.001.

Abbreviations: BMI, body mass index; CI, Confidence Interval; FN, false negative; FP, false positive; GNRI, Geriatric Nutritional Risk Index; MAC, mid-arm circumference (cm); MNA, Mini Nutritional Assessment; MNA-sf, Mini Nutritional Assessment-short form; MUST, Malnutrition Universal Screening Tool; NPV, negative predictive value; NR, nutritional risk, NRI, Nutritional Risk Index; NRS-2002, Nutritional Risk Screening tool 2002; NUTRIC, Nutrition Risk in the Critically Ill (NUTRIC) scores; PPV, positive predictive value; TLC, total lymphocyte count; TN, true negative; TP, true positive; TSFT, triceps skin-fold thickness (mm).

**Table 6 nutrients-12-02956-t006:** Predictive validity of various tools used to evaluate nutritional risk or malnutrition in older adults with COVID-19.

Author	NST	Length of Stay (LOS)	Appetite Change	Weight Change	Hospital Expenses	Complications	Mortality
Liu et al., 2020 [46]	NRS-2002	Nutritional risk predicted longer LOS; OR (95% CI): 0.102 (0.042–0.250), *p* = 0.000; AUC for LOS > 30 days (95% CI): 0.724 (0.640–0.808), *p* = 0.000. Rating: Weak.	Nutritional risk predicted change in appetite; OR (95% CI) for no change: 11.179 (3.881–32.169), *p* = 0.000; AUC for poor appetite (95% CI): 0.670 (0.586–0.747), *p* = 0.014. Rating: Poor.	Nutritional risk predicted weight change; OR (95% CI): 0.128 (0.047–0.350), *p* = 0.000; AUC for weight change >2.6 kg (95% CI): 0.613 (0.528–0.694), *p* = 0.000. Rating: Poor.	Nutritional risk predicted higher hospital expenses (CNY); OR (95% CI): 0.131 (0.054–0.313), *p* = 0.000; AUC for hospital expenses > CNY 56,163 (95% CI): 0.667 (0.583–0.744), *p* = 0.000. Rating: Poor.	Nutritional risk predicted greater disease severity; OR (95% CI): 0.095 (0.031–0.292), *p* = 0.000.	-
Liu et al., 2020 [46]	MNA-sf	Nutritional risk predicted longer LOS; OR (95% CI): 0.401 (0.198–0.813), *p* = 0.011; AUC for LOS > 30 days (95% CI): 0.602 (0.304–0.492), *p* = 0.032. Rating: Poor.	Nutritional risk predicted change in appetite; OR (95% CI) for no change: 40.731 (13.681–121.389), *p* = 0.000; AUC for poor appetite (95% CI): 0.868 (0.801–0.919), *p* = 0.000. Rating: Good.	Nutritional risk predicted weight change; OR (95% CI): 0.085 (0.035–0.206), *p* = 0.000; AUC for weight change >2.6 kg (95% CI): 0.895 (0.832–0.940), *p* = 0.000. Rating: Good.	Nutritional risk predicted higher hospital expenses (CNY); OR (95% CI): 0.436 (0.216–0.880), *p* = 0.021; AUC for hospital expenses > CNY 56,163 (95% CI): 0.597 (0.511–0.679), *p* = 0.063. Rating: Failure.	Nutritional risk predicted greater disease severity; OR (95% CI): 0.632 (0.289–1.382), *p* = 0.250. Rating: Poor.	-
Liu et al., 2020 [46]	MUST	Nutritional risk did not predict longer LOS; OR (95% CI): 0.722 (0.391–1.334), *p* = 0.298; non-significant AUC for LOS > 30 days (95% CI): 0.506 (0.421–0.591), *p* = 0.887.	Nutritional risk predicted change in appetite; OR (95%CI) for no change: 2.866 (1.449–5.669), *p* = 0.002; AUC for poor appetite (95% CI): 0.614 (0.528–0.694), *p* = 0.009. Rating: Poor.	Nutritional risk predicted weight change; OR (95% CI): 0.009 (0.003–0.026), *p* = 0.000; AUC for weight change >2.6 kg (95% CI): 0.887 (0.823–0.934), *p* = 0.000. Rating: Good.	Nutritional risk did not predict higher hospital expenses (CNY); OR (95% CI): 0.599 (0.323–1.109), *p* = 0.103; non-significant AUC for hospital expenses > CNY 56,163 (95% CI): 0.516 (0.430–0.601), *p* = 0.735.	Nutritional risk did not predict greater disease severity OR (95% CI): 1.367 (0.688–2.718), *p* = 0.372.	-
Liu et al., 2020 [46]	NRI	Nutritional risk predicted longer LOS; OR (95% CI): 0.261 (0.133–0.513), *p* = 0.000; AUC for LOS > 30 days (95% CI): 0.664 (−0.579 to 0.741), *p* = 0.000. Rating: Poor.	Nutritional risk predicted change in appetite; OR (95% CI) for no change: 2.768 (1.363–5.618). *p* = 0.005; AUC for poor appetite (95% CI): 0.629 (0.544–0.709), *p* = 0.014. Rating: Poor.	Nutritional risk predicted weight change; OR (95% CI): 0.182 (0.087–0.378), *p* = 0.000; AUC for weight change >2.6 kg (95% CI): 0.697 (0.614–0.772), *p* = 0.000. Rating: Poor.	Nutritional risk predicted higher hospital expenses (CNY); OR (95% CI): 0.199 (0.100–0.397), *p* = 0.000; AUC for hospital expenses > CNY 56,163 (95% CI): 0.621 (0.535–0.701), *p* = 0.019. Rating: Poor.	Nutritional risk predicted greater disease severity; OR (95% CI): 0.367 (0.173–0.776), *p* = 0.009.	-
Zhang et al., 2020 [43]	mNUTRIC score					Nutritional risk correlated with complications during ICU stay: ARDS (*p* < 0.001), shock (*p* < 0.001), acute myocardial injury (*p* = 0.002), and secondary infection (*p* = 0.002). Rating: Good. No correlation with acute liver dysfunction (*p* = 0.820), acute kidney injury (*p* = 0.172), embolization/thrombosis (*p* = 0.281), or pneumothorax (*p* = 0.856).	Nutritional risk correlated with death in the ICU after 28 days (*p* < 0.001). Rating: Good.

Abbreviations: ARDS, acute respiratory distress syndrome; AUC, Area Under the Curve; CI, Confidence Interval; CNY, Chinese Yuan; COVID-19, coronavirus disease 2019; ICU, intensive care unit; MNA-sf, Mini Nutritional Assessment-short form; mNUTRIC, modified Nutrition Risk in the Critically Ill; MUST, Malnutrition Universal Screening Tool; OR, Odds Ratio; NRI, Nutritional Risk Index; NRS-2002, Nutritional Risk Screening 2002; NST, Nutritional Screening Tool.

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
