# Peer review of "Nutritional Risk Screening Tools for Older Adults with COVID-19: A Systematic Review"

_nutrients, 2020, doi:10.3390/nu12102956_

Round 1

Reviewer 1 Report

The article is relevant and it´s well organized. However, some aspects of the introduction may be improved.

Introduction

The second paragraph only describes the role of inflammation in nutrition status and covid-19. More information regarding how nutrition status is related to the treatment and severity of covid19 should be explained, specially malnutrition (which is the main theme)

The  paragraph number 4 (line58-61) does not show any new information than the third one.

The introduction should improved with a better explanation of how nutritional status may be related with the infection and treatment of covid19 (including malnutrition)

Although the theme is extremely relevant, this relevance is not well exposed in the introduction.

Materials and methods

More details regarding what type of Observational study should be explained (cross-sectional, longitudinal, case control, or all?). The type of study described as “comparative” is case-control? Why not include RCT?

Were case studies included? It is not described in lines 77 to 75.

The criteria of each screening tool could be shown in a table. It would also reduce help to reduce the discussion.

Discussion

The discussion could be shorter and more concise. I suggest to focus on the prevalence of malnutrition in COVID19 patients and explaining the difference between studies and their criteria (tools used). In some parts of the discussion the authors give long explanation about pathophysiological aspects of covid19 (line 388 to 409), which is not the main aspect of this systematic review.

Author Response

Response to Reviewer 1 Comments

The article is relevant and it´s well organized. However, some aspects of the introduction may be improved.

Introduction

  1. The second paragraph only describes the role of inflammation in nutrition status and covid-19. More information regarding how nutrition status is related to the treatment and severity of covid19 should be explained, specially malnutrition (which is the main theme).

Answer:

Thank you for this observation. Inclusion performed according to request, please see the Introduction section, page 2, lines 45-51: “The course of COVID-19 presents with one severe inflammatory condition characterized by the involvement of proinflammatory cytokines [5]. Malnutrition may compromise the patient’s immune system and, consequently, the effectiveness of treatment, resulting in greater disease severity [6-7]. In this sense, the early assessment of nutritional risk, defined as “chances of a better or worse outcome from disease or surgery according to actual or potential nutritional and metabolic status” [8], can contribute to the determination of the most appropriate nutritional therapy. Thus, adequate nutrition can provide a better immune system response and a more favorable prognosis [6–7].

  1. The  paragraph number 4 (line58-61) does not show any new information than the third one.

Answer:

Thank you for this observation. The fourth paragraph has been summarized and has been incorporated with the third, please see the Introduction section, page 2, lines 52-64: “There is no gold standard for identifying nutritional risk or malnutrition. In most cases, nutritional risk is researched via screening tools, typically applied by nutritionists, doctors, or other professionals, preceding a full nutritional assessment [9,10]. Tools such as the Mini Nutritional Assessment-short form (MNA-sf) [11,12], the Geriatric Nutrition Risk Index (GNRI) [13], the Nutritional Risk Screening 2002 (NRS-2002) [14], the Malnutrition Universal Screening Tool (MUST) [15], the Nutritional Risk Index (NRI) [16], the Short Nutritional Assessment Questionnaire (SNAQ) [17], and Nutrition Risk in the Critically Ill (NUTRIC) scores [18] are practical and inexpensive to apply and can predict clinical outcomes [9]. Nutritional assessment instruments, such as the Global Assessment Subjective (SGA) and the Mini Nutritional Assessment (MNA), also assess biochemical and laboratory parameters and clinical and dietetical factors [9,10]. The instruments MNA [19,20], MNA-sf [11,12,21], GNRI [13,21], MUST [15,21], and NRI [16,21] were developed specifically to identify nutritional risk or malnutrition in older patients and other are also used to evaluate this target population [22-23].

  1. The introduction should improved with a better explanation of how nutritional status may be related with the infection and treatment of covid19 (including malnutrition)

Answer:

Thank you for this observation. Modification performed according to request, please see the Introduction section, page 2, lines 45-51: “The course of COVID-19 presents with one severe inflammatory condition characterized by the involvement of proinflammatory cytokines [5]. Malnutrition may compromise the patient’s immune system and, consequently, the effectiveness of treatment, resulting in greater disease severity [6-7]. In this sense, the early assessment of nutritional risk, defined as “chances of a better or worse outcome from disease or surgery according to actual or potential nutritional and metabolic status” [8], can contribute to the definition of the most appropriate nutritional therapy, Thus, adequate nutrition can provide to a better immune system response and a more favorable prognosis [6–7].” In addition, please see Introduction section, page 2, lines 65-69: “Identifying older patients at nutritional risk proves difficult in the intensive care unit (ICU) due to the nature of critical illness. No consensus exists on the most appropriate method to identify these patients. Traditional screening and assessment tools did not uniformly identify older patients as malnourished or at nutritional risk in the ICU and, therefore, may be inappropriate for use in this population [24]”. 

  1. Although the theme is extremely relevant, this relevance is not well exposed in the introduction.

Answer:

Thank you for this observation. The importance of assessing the nutritional risk in older patients with COVID-19 was highlighted in the second paragraph of the introduction, to check Introduction section, page 2, lines 45-51: “The course of COVID-19 presents with one severe inflammatory condition characterized by the involvement of proinflammatory cytokines [5]. Malnutrition may compromise the patient’s immune system and, consequently, the effectiveness of treatment, resulting in greater disease severity [6-7]. In this sense, the early assessment of nutritional risk, defined as “chances of a better or worse outcome from disease or surgery according to actual or potential nutritional and metabolic status” [8], can contribute to the definition of the most appropriate nutritional therapy, Thus, adequate nutrition can provide to a better immune system response and a more favorable prognosis [6–7]. In addition, the importance of nutritional screening was also highlighted in the fourth paragraph, which was included as suggested, please see the Introduction section, page 2, lines 65-69: “Identifying older patients at nutritional risk proves difficult in the intensive care unit (ICU) due to the nature of critical illness. No consensus exists on the most appropriate method to identify these patients. Traditional screening and assessment tools did not uniformly identify older patients as malnourished or at nutritional risk in the ICU and, therefore, may be inappropriate for use in this population [24]”. 

  1. Materials and methods

More details regarding what type of Observational study should be explained (cross-sectional, longitudinal, case control, or all?). The type of study described as “comparative” is case-control? Why not include RCT?

Answer:

Thank you for this observation. Modification performed according to request, please see the Material and Methods section, page 2, lines 83-86: “Observational studies (cross-sectional, longitudinal, case series, case-control), clinical trials, comparative studies of different nutritional risk screening instruments, or validation studies that used nutritional screening instruments to identify nutritional risk in patients over 65 years of age with COVID-19 were considered eligible for inclusion in the study. Review articles were excluded”.

Only review studies were defined as exclusion criteria. Randomized controlled trials were not excluded. No eligible RCTs were identified for inclusion in the systematic review. 

  1. Were case studies included? It is not described in lines 77 to 75.

Answer:

Thank you for this observation. Yes, please see the Material and Methods section, page 2, lines 83-86: “Observational studies (cross-sectional, longitudinal, case series, case-control), clinical trials, comparative studies of different nutritional risk screening instruments, or validation studies that used nutritional screening instruments to identify nutritional risk in patients over 65 years of age with COVID-19 were considered eligible for inclusion in the study. Review articles were excluded”. 

  1. The criteria of each screening tool could be shown in a table. It would also reduce help to reduce the discussion.

Answer:

Thank you for this observation. Inclusion performed according to request, please see the Table 4, page 9-10, lines 233-239.

  1. Discussion

The discussion could be shorter and more concise. I suggest to focus on the prevalence of malnutrition in COVID19 patients and explaining the difference between studies and their criteria (tools used). In some parts of the discussion the authors give long explanation about pathophysiological aspects of covid19 (line 388 to 409), which is not the main aspect of this systematic review.

Answer:

Thank you for this observation. The discussion was summarized as requested. Paragraphs 1 and 2 have been merged, please see the Discussion section, page 17, lines 301-311: “To our knowledge, this is the first systematic review to verify the nutritional screening tools used to identify nutritional risk in older patients with COVID-19. The instruments evaluated in this review can be considered useful for assessing nutritional risk in older COVID-19 patients. Although the majority (except the MUST) demonstrated low specificity, they showed high sensitivity and/or good predictive validity. The NRS-2002 was the only instrument to demonstrate a sensitivity of 100% compared to BMI and greater than 90% compared to three other nutritional screening instruments (the MNA-sf, MUST, and NRI). However, the MUST demonstrated better specificity. The MNA and the MUST showed better criterion validity. When tools with high sensitivity and specificity are not available, high sensitivity to identify risk is the preferable criterion for selecting a screening instrument because this is essential for directing individualized nutritional management with a focus on improving the patient’s nutritional status [47,48].

The exclusion of the two paragraphs (in the previous version the text was on lines 388-409) on the pathophysiological aspects of COVID-19 was performed as requested, please see the Discussion section.

Reviewer 2 Report

The manuscript is an interesting analysis of studies related to the assessment of malnutrition in the context of COVID-19. As such, it is a welcome contribution to the knowledge of the disease, since such papers are still in short supply. The analysis is performed thoroughly and carefully; the discussion is detailed and offers a multitude of clinical paths for a skilled nutritionist. Weaknesses of this review are pointed out in the text and result from a small number of qualified papers, small samples analyzed in these papers, and the geographical limitation to studies from China. Thereby, the proof strength and possibility of extrapolation are limited. Nonetheless, the risk of malnutrition is highly prevalent in COVID-19 patients, and incorrect nutrition may adversely affect the outcomes.

The discussion may be difficult to read by a general audience including primary physicians who are often the first medical specialists who must deal with a variety of aspects of COVID-19. Hence, this paper would benefit from a simplified algorithm for the selection of screening tools depending on the clinical status of the patient – possibly as part of Conclusions or a separate figure.

As for the Methods, the following sentences are very general and obvious. The authors could perhaps provide more specific information here.

45-47: The course of COVID-19 presents with one severe inflammatory condition characterized by the involvement of proinflammatory cytokines [5]. Thus, adequate nutrition can contribute to a better immune system response and a more favorable prognosis [6–8].

Next, the sentences below are unclear and should be further elaborated to reflect the actual process of the analyses performed:

112-114: One narrative summary of the nutritional screening tools used to identify nutritional risk in elderly patients with COVID-19 was conducted. Descriptive analysis was performed to synthesize the data on the instruments’ properties.

Generally, one should avoid the adjective "elderly" as it is nowadays often perceived as stigmatizing.  Possible alternatives include "older persons," "older people," "older adults" or "older subjects."

Author Response

Response to Reviewer 2 Comments

The manuscript is an interesting analysis of studies related to the assessment of malnutrition in the context of COVID-19. As such, it is a welcome contribution to the knowledge of the disease, since such papers are still in short supply. The analysis is performed thoroughly and carefully; the discussion is detailed and offers a multitude of clinical paths for a skilled nutritionist. Weaknesses of this review are pointed out in the text and result from a small number of qualified papers, small samples analyzed in these papers, and the geographical limitation to studies from China. Thereby, the proof strength and possibility of extrapolation are limited. Nonetheless, the risk of malnutrition is highly prevalent in COVID-19 patients, and incorrect nutrition may adversely affect the outcomes.

  1. The discussion may be difficult to read by a general audience including primary physicians who are often the first medical specialists who must deal with a variety of aspects of COVID-19. Hence, this paper would benefit from a simplified algorithm for the selection of screening tools depending on the clinical status of the patient – possibly as part of Conclusions or a separate figure.

Answer:

Thank you for this observation. Table 4 was included to show the evaluation parameters of the instruments and the setting recommendations for the application of each one, please see the Table 4, page 9-10, lines 233-239.

  1. As for the Methods, the following sentences are very general and obvious. The authors could perhaps provide more specific information here.

45-47: The course of COVID-19 presents with one severe inflammatory condition characterized by the involvement of proinflammatory cytokines [5]. Thus, adequate nutrition can contribute to a better immune system response and a more favorable prognosis [6–8].

Answer:

Thank you for this observation. Modification performed according to request, please see the Introduction section, page 2, lines 45-51: “The course of COVID-19 presents with one severe inflammatory condition characterized by the involvement of proinflammatory cytokines [5]. Malnutrition may compromise the patient’s immune system and, consequently, the effectiveness of treatment, resulting in greater disease severity [6-7]. In this sense, the early assessment of nutritional risk, defined as “chances of a better or worse outcome from disease or surgery according to actual or potential nutritional and metabolic status” [8], can contribute to the determination of the most appropriate nutritional therapy. Thus, adequate nutrition can provide a better immune system response and a more favorable prognosis [6–7]. 

  1. Next, the sentences below are unclear and should be further elaborated to reflect the actual process of the analyses performed:

112-114: One narrative summary of the nutritional screening tools used to identify nutritional risk in elderly patients with COVID-19 was conducted. Descriptive analysis was performed to synthesize the data on the instruments’ properties.

Answer:

Thank you for this observation. Modification performed according to request, please see the Material and Methods section, page 3, lines 121-125: “Descriptive analysis was performed to synthesize the data on the instruments’ properties (criterion validity, construct validity, and predictive validity) analyzed in each study. One narrative synthesis comparing the same properties between the different nutritional risk screening instruments was conducted. The prevalence of nutritional risk and malnutrition was compared between studies, considering the instruments used in each one”.

  1. Generally, one should avoid the adjective “elderly” as it is nowadays often perceived as stigmatizing.  Possible alternatives include “older persons,” “older people,” “older adults” or “older subjects.”

Answer:

Thank you for this observation. Throughout the article, the word “elderly” has been replaced by “older” (highlighted in red) to address this comment.
